# GENERALING MULTIMODAL VARIATIONAL METHODS TO SETS

## ABSTRACT

Making sense of multiple modalities can yield a more comprehensive description of real-world phenomena. However, learning the co-representation of diverse modalities is still a long-standing endeavor in emerging machine learning applications and research. Previous generative approaches for multimodal input approximate a joint-modality posterior by uni-modality posteriors as product-of-experts (PoE) or mixture-of-experts (MoE). We argue that these approximations lead to a defective bound for the optimization process and loss of semantic connection among modalities. This paper presents a novel variational method on sets called the Set Multimodal VAE (SMVAE) for learning a multimodal latent space while handling the missing modality problem. By modeling the joint-modality posterior distribution directly, the proposed SMVAE learns to exchange information between multiple modalities and compensate for the drawbacks caused by factorization. In public datasets of various domains, the experimental results demonstrate that the proposed method is applicable to order-agnostic cross-modal generation while achieving outstanding performance compared to the state-of-the-art multimodal methods. The source code for our method is available online https://anonymous.4open.science/r/SMVAE-9B3C/.

## 1 INTRODUCTION

Most real-life applications such as robotic systems, social media mining, and recommendation systems naturally contain multiple data sources, which raise the need for learning co-representation among diverse modalities Lee et al. (2020). Making use of additional modalities should improve the general performance of downstream tasks as it can provide more information from another perspective. In literatures, substantial improvements can be achieved by utilizing another modality as supplementary information Asano et al. (2020); Nagrani et al. (2020) or by multimodal fusion Atrey et al. (2010); Hori et al. (2017); Zhang et al. (2021). However, current multimodal research suffers severely from the lack of multimodal data with fine-grained labeling and alignment Sun et al. (2017); Beyer et al. (2020); Rahate et al. (2022); Baltrušaitis et al. (2018) and the missing of modalities Ma et al. (2021); Chen et al. (2021).

In the self-supervised and weakly-supervised learning field, the variational autoencoders (VAEs) for multimodal data Kingma & Welling (2013); Wu & Goodman (2018); Shi et al. (2019); Sutter et al. (2021) have been a dominating branch of development. VAEs are generative self-supervised models by definition that capture the dependency between an unobserved latent variable and the input observation. To jointly infer the latent representation and reconstruct the observations properly, the multimodal VAEs are required to extract both modality-specific and modality-invariant features from the multimodal observations. Earlier works mainly suffer from scalability issues as they need to learn a separate model for each modal combination Pandey & Dukkipati (2017); Yan et al. (2016). More recent multimodal VAEs handle this issue and achieves scalability by approximating the true joint posterior distribution with the mixture or the product of uni-modality inference models Shi et al. (2019); Wu & Goodman (2018); Sutter et al. (2021). However, our key insight is that their methods suffer from two critical drawbacks: 1) The implied conditional independence assumption and corresponding factorization deviate their VAEs from modeling inter-modality correlations. 2) The aggregation of inference results from uni-modality is by no means a co-representation of these modalities.

To overcome these drawbacks of previous VAE methods, this work proposes the Set Multimodal Variational Autoencoder (SMVAE), a novel multimodel generative model eschewing factorization and instead relying solely upon set operation to achieve scalability. The SMVAE allows for better performance compared to the latest multimodal VAE methods and can handle input modalities of variable numbers and permutations. By learning the actual multimodal joint posterior directly, the SMVAE is the first multimodal VAE method that achieves scalable co-representation with missing modalities. A high-level overview of the proposed method is illustrated in Fig.1. The SMVAE can handle a set of maximally $M$ modalities as well as their subsets and allows cross-modality generations. $E_i$ and $D_i$ represent the $i-$th embedding network and decoder network for the specific modality. $\mu_s, \sigma_s$ and $\mu_k, \sigma_k$ represent the parameters for the posterior distribution of the latent variable. By incorporating set operation when learning the joint-modality posterior, we can simply drop the corresponding embedding networks when a modality is missing. Comprehensive experiments show the proposed Set Multimodal Variational Autoencoder (SMVAE) outperforms state-of-the-art multimodal VAE methods and is immediately applicable to real-life multimodality.

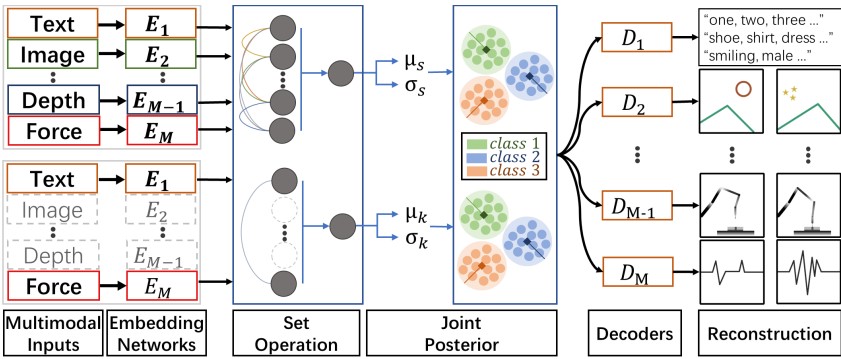

Figure 1: Overview of the proposed method for learning multimodal latent space. The SMVAE is able to handle any combination or number of input modalities while having discriminative latent space and proper reconstruction.

## 2 RELATED WORK

### 2.1 MULTIMODALITY VAEs

The core problem of learning a multimodal generative model is to maintain the model's scalability to the exponential number of modal combinations. Existing multimodal generative models such as Conditional VAE (CVAE)Pandey & Dukkipati (2017) and joint-modality VAE (JMVAE) Suzuki et al. (2016) had difficulty scaling since they need to assign a separate inference model for each possible input and output combinations. To tackle this issue, follow-up works, such as, TELBO Vedantam et al. (2017), MVAE Wu & Goodman (2018), MMVAE Shi et al. (2019), MoPoE Sutter et al. (2021), assume the variational approximation is factorizable. Thus, they focused on factorizing the approximation of the multimodal joint posterior $q(\mathbf{z}|\mathbf{x}_1, \cdots, \mathbf{x}_M)$ into a set of uni-modality inference encoders $q_i(\mathbf{z}|\mathbf{x}_i)$, such that $q(\mathbf{z}|\mathbf{x}_1, \cdots, \mathbf{x}_M) \approx F(\{\mathbf{x}_i\}_{i=1}^M)$, where $F(\cdot)$ is a product or mean operation, depending on the chosen aggregation method. As discussed in Sutter et al. (2021), these scalable multimodal VAE methods differ only in the choice of aggregation method. Different from those mentioned above multimodal VAE methods, we attain the joint posterior in its original form without introducing additional assumptions on the form of the joint posterior. To handle the issue of scalability, we exploit the deterministic set operation function in the noise-outsourcing process. While existing multimodal VAE methods can be viewed as typical late fusion method that combines decisions about the latent variables Khaleghi et al. (2013), the proposed SMVAE method corresponds to the early fusion method at the representation level, allowing for the learning of correlation and co-representation from multimodal data.

## 2.2 Methods for set-input problems

Multiple instance learning (MIL) Carbonneau et al. (2018) and 3D shape recognition Su et al. (2015); Hofer et al. (2005); Wu et al. (2015), are well-known examples of weakly-supervised learning problems that deal with set-input. MIL handles training data as numerous sets of instances with only set-level labels. A typical way to solve set-level classification problems is to use pooling methods for information aggregation Shao et al. (2021). Recently, Lee et al. (2019) observed that classical feed-forward neural networks like the multi-layer perception (MLP) Murtagh (1991) cannot guarantee invariance under the permutation of the elements in the input as well as the input of arbitrary sizes. Furthermore, recursive neural networks such as RNN and LSTM Hochreiter & Schmidhuber (1997) are sensitive to the order of the input sequences, and cannot fit the multimodal case since there is no natural order for modalities. Recently, Deep Sets Zaheer et al. (2017) provided a formal definition for a permutation invariant function in set-input problems and proposed a universal approximator for arbitrary set functions. Later on, Set Transformer Lee et al. (2019) further extends this idea by using the self-attention mechanism to provide interactions as well as information aggregation among elements from an input set. However, their method only models a set of outputs as a deterministic function. Our work fills the gap between a deterministic set function to a probabilistic distribution and applies it to multimodal unsupervised learning.

## 3 Proposed Method

### 3.1 Preliminaries

This work considers the multimodal learning problem as a set modeling problem and presents a scalable method for learning multimodal latent variables and cross-modality generation. Given a dataset $\{\mathbb{X}^{(i)}\}_{i=1}^{N}$ of $N$ i.i.d. multimodal samples, we consider each of the sample as a set of $M$ modalities observations $\mathbb{X}^{(i)} = \{\mathbf{x}_j^{(i)}\}_{j=1}^{M}$. The multimodal data is assumed to be generated following the successive random process $p(\mathbb{X}, \mathbf{z}) = p_\theta(\mathbb{X}|\mathbf{z})p(\mathbf{z})$ which involves an unobserved latent variable $\mathbf{z}$. The prior distribution of the latent variable $\mathbf{z}$ is assumed to be $p_\theta(\mathbf{z})$, with $\theta$ denoting its parameters. The marginal log-likelihood of this dataset of multimodal sets can be expressed as a summation of marginal log-likelihood of individual sets as $\log p(\mathbb{X}^{(i)})$ as $\log \prod_{i=1}^{N} p(\mathbb{X}^{(i)}) = \sum_{i=1}^{N} \log p(\mathbb{X}^{(i)})$. Since the marginal likelihood of the dataset is intractable, we cannot optimize $p(\{\mathbb{X}^{(i)}\}_{i=1}^{N})$ with regards to $\theta$ directly. We instead introduce the variational approximation $q_\phi(\mathbf{z}|\mathbb{X})$ from a parametric family, parameterized by $\phi$, as an importance distribution. $q_\phi(\mathbf{z}|\mathbb{X})$ is often parameterized by a neural network with $\phi$ as its trainable parameters. Together, we can express the marginal log-likelihood of a single multimodal set as:

$$\log p(\mathbb{X}^{(i)}) = D_{KL}(q_\phi(\mathbf{z}|\mathbb{X}^{(i)})||p_\theta(\mathbf{z}|\mathbb{X}^{(i)})) + \mathcal{L}(\phi, \theta; \mathbb{X}^{(i)})$$

$$\mathcal{L}(\phi, \theta; \mathbb{X}^{(i)}) = \mathbb{E}_{\mathbf{z}\sim q_\phi(\mathbf{z}|\mathbb{X}_{(i)})}\left[\log p_\theta(\mathbb{X}^{(i)}), \mathbf{z}) - \log q_\phi(\mathbf{z}|\mathbb{X}^{(i)})\right] \quad (1)$$

$$= -D_{KL}(q_\phi(\mathbf{z}|\mathbb{X}^{(i)})||p_\theta(\mathbf{z})) + \mathbb{E}_{\mathbf{z}\sim q_\phi}(\mathbf{z}|\mathbb{X}^{(i)})\left[\log p_\theta(\mathbb{X}^{(i)}|\mathbf{z})\right]$$

, where $D_{KL}(\cdot||\cdot)$ is the Kullback-Leibler (KL) divergence between two distributions. The non-negative property of the KL divergence term between the variational approximation $q_\phi(\mathbf{z}|\mathbb{X}^{(i)})$ and the true posterior $p_\theta(\mathbf{z}|\mathbb{X}^{(i)})$ in the first line makes $\mathcal{L}(\phi, \theta; \mathbb{X}^{(i)})$ the natural evidence lower bound (ELBO) for the marginal log-likelihood. The last line indicates that maximizing the ELBO is equivalent to maximizing the reconstruction performance and regulating the variational approximation using the assumed prior distribution for the latent variable. To avoid confusion, we term neural networks used for mapping the raw input observations into a fixed-sized feature vector as the embedding network while the neural network used to parameterize the variational approximation $q_\phi(\mathbf{z}|\mathbb{X}_{(i)})$ as the encoder network. A frequently used version of the objective function is written as:

$$\arg \min_{\phi} - \beta D_{KL}(q_\phi(\mathbf{z}|\mathbb{X}^{(i)})||p(\mathbf{z})) + \mathbb{E}_{\mathbf{z}\sim q_\phi(\mathbf{z}|\mathbb{X}^{(i)})}\left[\lambda \log p(\mathbb{X}^{(i)}|\mathbf{z})\right] \quad (2)$$

, where additional annealing coefficients $\beta$ and reweighting coefficient $\lambda$ are used in the ELBO to allow gradients and warm-up training which gradually increases the regularization effect from the prior distribution and avoids reaching local minima in the early training stage Bowman et al. (2015); Sønderby et al. (2016). We drop the superscript of $\mathbb{X}^{(i)}$ to maintain brevity in the following paper.

## 3.2 SET MULTIMODAL VARIATIONAL AUTOENCODER

In multimodal scenarios with missing modalities, we consider each sample $\mathbb{X}_s = \{\mathbf{x}_i | i^{th} \text{modaltiy}$ present$\}$ as a subset of $\mathbb{X}$ and the powerset $\mathcal{P}(\mathbb{X})$ denoting all the $2^M$ combinations, such that $\mathbb{X}_s \in \mathcal{P}(\mathbb{X})$. Our goal is to perform inference and generation from any number and permutation of available modalities, which requires an inference process is invariant to permutations and input of variable size. Following Definition 1, we denotes the invariant inference process as $p(z|\mathbb{X}_s) = p(z|\pi \cdot \mathbb{X}_s)$. The ELBO for a subset $\mathbb{X}_s$ can be written as Eq.3.

$$\mathcal{L}_s(\phi, \theta; \mathbb{X}_s) = -D_{KL}(q_\phi(\mathbf{z}|\mathbb{X}_s)||p_\theta(\mathbf{z})) + \mathbb{E}_{\mathbf{z} \sim q_\phi}(\mathbf{z}|\mathbb{X}_s)[\log p_\theta(\mathbb{X}_s|\mathbf{z})] \tag{3}$$

**Definition 1** *Let $S_n$ be a set of all permutations of indices $1, \cdots, N$, $X = (\mathbf{x}_1, \cdots \mathbf{x}_n)$ denotes $n$ random variables. A probabilistic distribution $p(y|X)$ is permutation inariant if and only if for any permutation $\pi \in S_n$, $p(y|X) = p(y|\pi \cdot X)$, where $\cdot$ is the group action.*

The difference between $\mathcal{L}(\phi, \theta; \mathbb{X})$ in Eq.1 and $\mathcal{L}_s(\phi, \theta; \mathbb{X}_s)$ in Eq.3 is that the ELBO for a subset $\mathbb{X}_s$ is not yet a valid bound for $\log p(\mathbb{X})$ by itself. Additional sampling from $\mathcal{P}(\mathbb{X})$ in the optimization objective as Eq.4 is needed for theoretical completeness.

$$\arg\min_\phi \sum_{\substack{\mathbb{X}_s \sim \mathcal{P}(\mathbb{X}) \\ \pi \in S_n}} \mathcal{L}_s(\phi, \theta; \pi \cdot \mathbb{X}_s) \tag{4}$$

, where $\pi$ is a randomly generated permutation to the input subset $\mathbb{X}_s$. However, this sampling process can be trivial if we combine the sampling of the subsets with the sampling of mini-batch during training. By assuming the Gaussian form of the latent variable $\mathbf{z}$ and applying the reparameterization technique, the inference process of SMVAE can be written as:

$$p(\mathbf{z}|\mathbf{x}_s) \sim \mathcal{N}(\mu, \sigma^2), \epsilon \sim \mathcal{N}(0, I) \tag{5}$$

$$\mathbf{z} := \mu + \sigma \odot \epsilon \tag{6}$$

$$\mu_z, \log \sigma_z^2 := g_\phi(E_1(\mathbf{x}_1), \cdots, E_m(\mathbf{x}_m)) \tag{7}$$

, where $E_i$ are embedding network for the $i^{th}$ modality, $g_\phi(\cdot)$ is a neural network with trainable parameters $\phi$ that provide the parameter for the latent's posterior distribution (i.e., $\mu$ and $\sigma$) , $\odot$ denotes the element-wise multiplication. For the generation process, it is desired to models the joint likelihood of modalities conditioned on the latent variables $p_\theta(\mathbf{x}_s, \mathbf{z}) = p(\mathbf{z})p_\theta(\mathbf{x}_s|\mathbf{z})$ so that the model can utilize information from other available modalities more easier when generating a complex modality. However, for the sake of easy implementation, we assign $n$ separate decoders $D_1, \cdots, D_M$ for all possible modalities as $p_\theta(\mathbf{x}_s|\mathbf{z}) = [D_{\theta_1}(z), \cdots, D_{\theta_M}(z)]$. We find empirically that, without loss of generality, using $L_2-$normalization as additional regularization to regulate the parameter o$\mu$ and $\sigma$ of the inference network to 0 and 1 respectively could facilitate the learning efficiency because the gradient from the ELBO often favors the reconstruction term over the regularization term.

## 3.3 SET REPRESENTATION FOR JOINT DISTRIBUTION

The scalability issue comes from the requirement for an inference process for the powerset $\mathcal{P}(\mathbb{X})$. We achieve scalability by using the noise-outsourced functional representation, i.e. $\mathbf{z} = g(\epsilon, \mathbb{X}_s)$, to bridge the gap between the deterministic set functions to a stochastic function. The properties of the deterministic function thus can be passed to the stochastic distribution under minor conditions Bloem-Reddy & Teh (2020). With such a foundation, the problem of modeling the posterior for a superset immediately reduces to designing a differentiable deterministic function that has the desired invariant or elastic properties. Specifically, we identify four critical requirements for weakly-supervised multimodal learning. Being that the model should 1) be scalable in the number of observable modalities; 2) be able to process input modalities sets of arbitrary size and permutation; 3) satisfy Theorem 1; and 4) be able to learn the co-representation among all modalities.

**Theorem 1** *A valid set function $f(x)$ is invariant to the permutation of instances, iif it can be decomposed in the form $\Phi(\sum \Psi(x))$, for any suitable transformations $\Phi$ and $\Psi$.*

An oversimplified example of a set function can be summation or product as done in MVAE Wu & Goodman (2018) and MMVAE Shi et al. (2019). Pooling operations such as average pooling or max pooling also fit the definition. However, these set aggregation operations will require additional factorization assumptions to the joint posterior and ultimately forbid the VAE to learn co-representation of the input modalities as aggregation is only applied at the decision level. To establish the inductive bias of inter-modality correlation, the self-attention mechanism without positional embeddings is a reasonable choice Edelman et al. (2022); Shvetsova et al. (2022).

Therefore, the proposed SMVAE leverages self-attention as the deterministic set function to aggregate embeddings of multimodal inputs. Given the query $Q$, key $K$ and value $V$, an attention function is denoted as $\text{Att}(Q, K, V) = \omega(\frac{QK^T}{\sqrt{d_k}})V$, where $K \in \mathbb{R}^{m \times d_k}$ and $V \in \mathbb{R}^{m \times d_v}$ are $m$ vectors of dimension $d_k$ and $d_v$, $Q \in \mathbb{R}^{n \times d_q}$ are $n$ vectors of dimension $d_q$, $\omega$ is the softmax activation function. In our case, the key-value pairs represent the $m$ available embeddings of input modalities, $m \leq M$. Each embedding is mapped to a $d-$dimensional embedding space by a modality-specific embedding network. By measuring the compatibility of the corresponding key and the query $Q$, information that is shared among modalities is aggregated as co-representation.

In practice, we utilize the multi-head extension of self-attention denoted as $\text{MultiHead}(Q, K, V, h) = \text{Concat}(A_1, \cdots, A_h)W^o$, where $A_i = \text{Att}_i(QW_i^Q, KW_i^K, VW_i^v)$ is obtained from the $i^{th}$ attention function with projection parameters $W_i^Q \in \mathbb{R}^{(d/h) \times d_q}, W_i^K \in \mathbb{R}^{(d/h) \times d_k}$, $W_i^V \in \mathbb{R}^{(d/h) \times d_k}$ and $W^o \in \mathbb{R}^{d_v \times d}$, $h$ denotes the total number of attention heads and $d$ denotes the dimension of the projections for keys, values and queries. Inspired by Lee et al. (2019), we design our deterministic set representation function $g_\phi(\mathbb{X}_s)$ as follows:

$$g_\phi(\mathbb{X}_s) := H + f_s(H)$$
$$H = I + \text{MultiHead}(I, \mathbb{X}_s, \mathbb{X}_s, h) \tag{8}$$

, where $I \in R^{1 \times d_v}$ is an $d_v$-dimensional trainable vector as the query vector for multimodal embeddings. $f_s$ is a fully-connected layer. By calculating attention weights using $I$ and each embedding. Not only does $I$ work as an aggregation vector that regulates the number of output vectors from $g_\phi(\mathbb{X}_s)$ to be constant regardless of the number of input embeddings, but also it selects relevant information from each embedding base on similarity measurement. The former justifies $g_\phi(\mathbb{X}_s)$ as a suitable permutation invariant set-processing function while the latter yields the desired co-representation among modalities. Finally, Since the set representation function $g_\phi(\mathbb{X}_s)$ is invariant to the input permutations of different input sizes, we achieved an invariant inference probabilistic function that satisfies Definition 1 through the noise-outsourced process as shown in Eq. 6. Thus, by introducing the set representation function in the noise-outsourced process, the SMVAE is readily a scalable multimodal model for any subsets of modalities.

## 3.4 TOTAL CORRELATION OPTIMIZATION WITHOUT CONDITION INDEPENDENCE

The lower bound of the multimodal data without factorizing the joint posterior (i.e., Eq. 1) provides additional information about the correlations of modalities during the optimization process compared to factorized methods. It is noteworthy that both MVAE and MMVAE depend on the assumption of conditional independence between modalities in factorization. Without loss of generality, the relation between $\mathcal{L}(\phi, \theta; \mathbb{X})$ and the factorized case $\mathcal{L}_{CI}$ can be shown in Eq. 9.

$$\mathcal{L}(\phi, \theta; \mathbb{X}) = \mathbb{E}_{q_\phi(\mathbf{z}|\mathbb{X})} \left[ \log \frac{p_\theta(z) \prod_{i=1}^M p_\theta(x_i \mid z)}{q_\phi(\mathbf{z} \mid \mathbb{X})} + \log \frac{p_\theta(\mathbb{X}, \mathbf{z})}{p_\theta(z) \prod_{i=1}^M p(x_i \mid z)} \right]$$
$$= \mathcal{L}_{CI} + \mathbb{E}_{q_\phi(\mathbf{z}|\mathbb{X})} \left[ \log \frac{p_\theta(\mathbb{X} \mid \mathbf{z})}{\prod_{i=1}^M p_\theta(\mathbf{x}_i \mid z)} \right] \tag{9}$$

, where $\mathbb{X} \equiv (\mathbf{x}_1, \cdots, \mathbf{x}_M)$ and $\mathcal{L}_{CI}$ is the lower bound for factorizable generative process as MVAE or MMVAE. Specifically, let $q(\mathbb{X})$ denotes the empirical distribution for the multimodal dataset, we

have:

$$\mathbb{E}_{q(\mathbb{X})}\left[\mathcal{L}(\phi,\theta;\mathbb{X})\right] = \mathbb{E}_{q(\mathbb{X})}\left[\mathcal{L}_{CI}\right] + \mathbb{E}_{z \sim \frac{q(\mathbb{X})q_\phi(z|\mathbb{X})}{p_\theta(\mathbb{X}|z)}}\left[\underbrace{\mathbb{E}_{\mathbb{X} \sim p_\theta(\mathbb{X}|\mathbf{z})}\left[\log\frac{p_\theta(\mathbb{X} \mid \mathbf{z})}{\prod_{i=1}^{M} p_\theta(x_i \mid \mathbf{z})}\right]}_{\text{conditional total correlation}}\right] \quad (10)$$

, which reveals that without assuming a factorizable generative process and enforcing conditional independence among modalities, our optimization objective naturally models the conditional total correlation which provides information of dependency among multiple input modalities Watanabe (1960); Studenỳ & Vejnarová (1998). Therefore, the SMVAE has the additional advantage of learning correlations among different modalities of the same event, which is also what we desired for good co-representation.

# 4 EXPERIMENTS

## 4.1 EXPERIMENT SETTINGS

We make use of uni-modal datasets including MNIST LeCun et al. (1998), FashionMNIST Xiao et al. (2017) and CelebA Liu et al. (2015) to evaluate the performance of the proposed SMVAE and compare with other state-of-the-art methods. We convert these uni-modal datasets into bi-modal dataset by transforming the labels to one-hot vectors as the second modality as in Wu & Goodman (2018); Suzuki et al. (2016). For quatitative evaluation, we denote $\mathbf{x}_1$ and $\mathbf{x}_2$ as the image and text modality and measure the marginal log-likelihood, $\log p(\mathbf{x}) \approx \log E_{q(\mathbf{z}|\cdot)}\left[\frac{p(\mathbf{x}|\mathbf{z})p(\mathbf{z})}{q(\mathbf{z}|\cdot)}\right]$, the joint likelihood $\log p(\mathbf{x}, \mathbf{y}) \approx \log E_{q(\mathbf{z}|\cdot)}\left[\frac{p(\mathbf{z})p(\mathbf{x}|\mathbf{z})p(\mathbf{y}|\mathbf{z})}{q(\mathbf{z}|\cdot)}\right]$, and the marginal conditional probability, $\log p(\mathbf{x}|\mathbf{y}) \approx \log E_{q(\mathbf{z}|\cdot)}\left[\frac{p(\mathbf{z})p(\mathbf{x}|\mathbf{z})p(\mathbf{y}|\mathbf{z})}{q(\mathbf{z}|\cdot)}\right] - \log E_{p(\mathbf{z})}\left[p(\mathbf{y}|\mathbf{z})\right]$, using data samples from the test set. $q(\mathbf{z}|\cdot)$ denotes the importance distribution. For all the multimodal VAE methods, we keep the architecture of encoders and decoders consistent for a fair comparison. Detailed training configurations and settings of the networks are listed in Appendix. B. The marginal probabilities measure the model's ability to capture data distributions while the conditional log probability measures classification performance. Higher scoring of these matrics means the better a model is able to generate proper samples and convert between modalities. These are the desirable properties for learning a generative model.

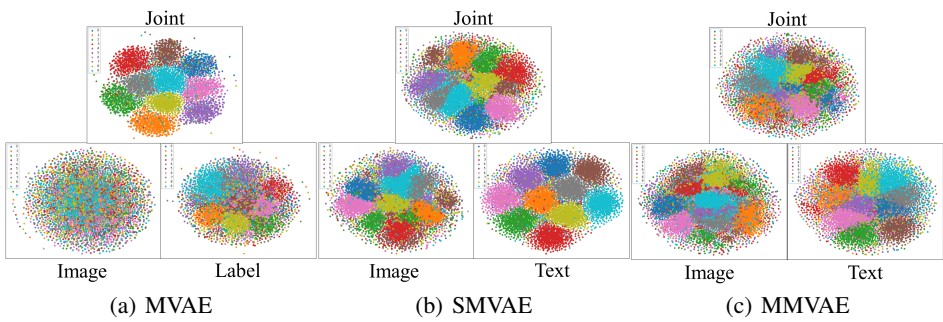

Figure 2: Visualization of 2-D latent representation learned by our model compared to MVAE and MMVAE from bi-modality MNIST dataset. Our model can learn natrual clusters while maintaining good structure for their joint distribution. Left: MVAE. Middle: SMVAE. Right: MMVAE.

## 4.2 GENERATION QUALITY AND QUANTITATIVE EVALUATION

We obtain 1000 important samples to estimate the probability metrics. Table 1 shows the quatitative results of the proposed SMVAE for each dataset. We can see that the SMVAE outperforms other methods in almost all metrics. The outstanding of SMVAE mainly contributes to the direct modeling of the joint posterior distribution and optimization on a more informative objective. Fig. 3, Fig. 4 and Fig. 5 show cross-modality generation of image samples for each domain generated by the SMVAE

Table 1: Statistical results

| DataSet | Method | sampled from $p(z\|x,y)$ | | | sampled from $p(z\|x)$ | | |
|---|---|---|---|---|---|---|---|
| | | $\log p(x,y)$ | $\log p(x)$ | $\log p(x\|y)$ | $\log p(x,y)$ | $\log p(x)$ | $\log p(x\|y)$ |
| FASHION | SMVAE | **-225.10** | **-230.81** | -232.85 | **-225.14** | **-232.36** | -232.84 |
| | JMVAE | -232.70 | -232.40 | **-230.65** | -232.94 | -232.63 | **-230.39** |
| | MVAE | -233.01 | -232.54 | -230.69 | -233.01 | -232.53 | -230.69 |
| | MMVAE | -235.37 | -233.16 | -231.32 | -231.25 | -235.18 | -237.64 |
| MNIST | SMVAE | **-83.41** | **-89.38** | **-86.45** | **-83.46** | **-89.09** | **-86.47** |
| | JMVAE | -91.03 | -90.962 | -88.43 | -90.76 | -90.69 | -88.69 |
| | MVAE | -90.85 | -90.616 | -88.55 | -90.85 | -90.61 | -88.56 |
| | MMVAE | -91.53 | -91.23 | -90.64 | -91.75 | -91.11 | -90.01 |
| | PoMoE | -90.16 | -90.29 | - | -91.36 | -91.77 | - |
| CelebA | SMVAE | **-6000.95** | **-6087.91** | **-6085.31** | **-6025.38** | **-6080.86** | **-6084.94** |
| | JMVAE | -6238.28 | -6234.54 | -6235.33 | -6242.19 | -6237.97 | -6231.46 |
| | MVAE | -6241.62 | -6237.10 | -6235.36 | -6242.03 | -6236.92 | -6234.95 |
| | MMVAE | -6513.43 | -6560.57 | -6516.14 | -6255.45 | -6284.28 | -6290.74 |

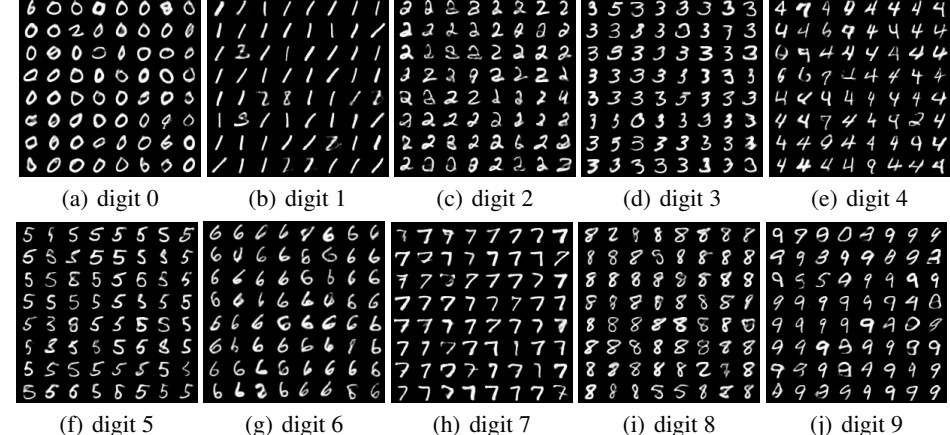

(a) digit 0    (b) digit 1    (c) digit 2    (d) digit 3    (e) digit 4

(f) digit 5    (g) digit 6    (h) digit 7    (i) digit 8    (j) digit 9

Figure 3: Conditional image reconstruction of digits generated by SMVAE given text input.

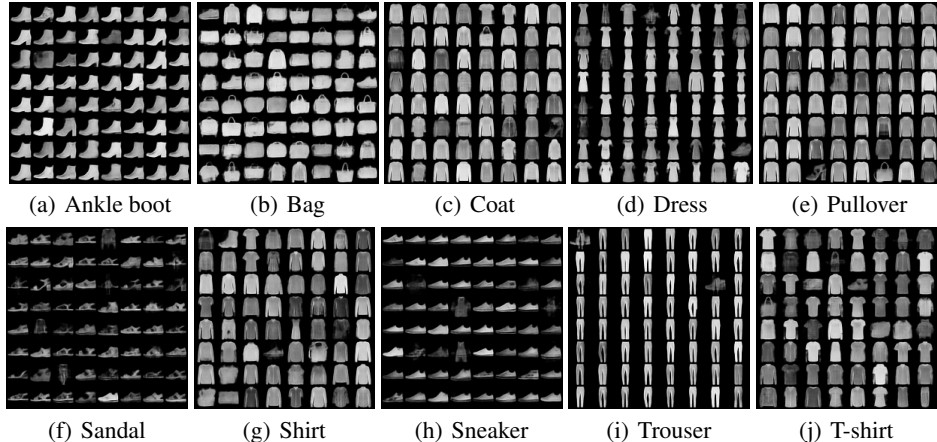

(a) Ankle boot    (b) Bag    (c) Coat    (d) Dress    (e) Pullover

(f) Sandal    (g) Shirt    (h) Sneaker    (i) Trouser    (j) T-shirt

Figure 4: Conditional image reconstruction of fashionMNIST generated by SMVAE given text input.

model. We can see that given the text modality only, the SMVAE can generate corresponding images of good quality.

We further visualize the learned latent representation using tSNE Hinton & Roweis (2002). As shown in Fig.2, latent space learned by MVAE method can only produce cohesive latent representation when

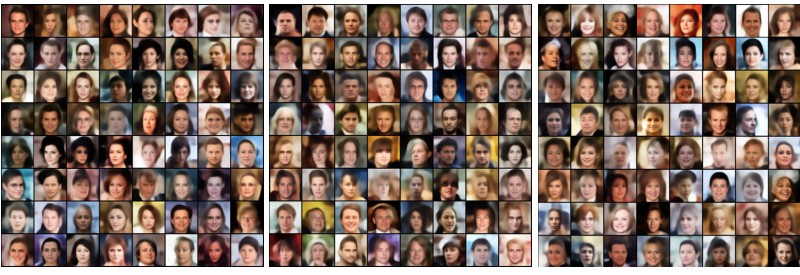

(a) Black Hair        (b) Male        (c) Smiling

Figure 5: Generated face images by SMVAE given facial attributes.

both modalities are presented. When one modality is missing, representations from their method are distributed irrespective of the semantic category of the data. On the other hand, although the MMVAE method achieves cohesive representation for single-modality posterior, their joint representation is less discriminative. Indicating that using only the combination of uni-modal inference networks is insufficient to capture intermodality co-representation. Nonetheless, our SMVAE method can achieve discriminative latent space for both single- and joint-modality inputs thanks to its ability to exploit shared information from different modalities.

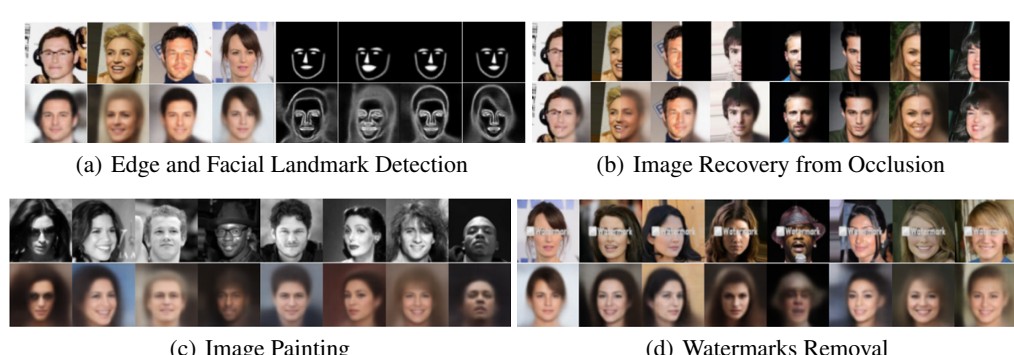

(a) Edge and Facial Landmark Detection        (b) Image Recovery from Occlusion

(c) Image Painting        (d) Watermarks Removal

Figure 6: Learning computer vision transformations: (a) Ground truth images are randomly selecetd from CelebA dataset and we present the generated recontructed images, detected edges and facial landmarks. (b) Recontructed color images based on grayscale images. (c) Image recovery from obscured image. (d) Image recovery from watermark images.

## 4.3 CASE STUDY: COMPUTER VISION APPLICATION

We demonstrate that our SMVAE is able to learn image transformations including colorization, edge detection, facial landmark segmentation, image completion, and watermark removal. With original image and each transformation as different modalities, we obtain 6 modalities in total by applying different transformations to the ground-truth images for this multimodal setting. This case study demonstrates the SMVAE's ability to generate in multiple directions and combinations.

Similar to Wu & Goodman (2018), for edge detection, we use Canny detector Canny (1986) from Scikit-Image module Van der Walt et al. (2014) to extract edges of the facial image. For facial landmark segmentation, we use Dlib tool King (2009) and OpenCV Bradski & Kaehler (2000). For colorization, we simply convert RGB colors to grayscale. For watermark removal, we add a watermark overlay to the original image. For image completion, we replace half of the image with black pixels. Fig.6 shows the samples generated from a trained SMVAE model. As can be seen in Fig.6(a), the SMVAE generates a good reconstruction of the facial landmark segmentation and extracted edges. In Fig.6(b), we can see that the SMVAE is able to put reasonable facial color to the input grayscale image. Fig.6(c) demonstrates that the SMVAE can recover the image from the watermark and complete the image quite well. The reconstructed right half of the image is basically agreed on the left half of the original image. In Fig.6(d), all traces of the watermark is also removed.

Although our reconstructed images suffer from the same blurriness problem that is shared in VAE methods Zhao et al. (2017), the SMVAE is able to perform cross-modality generation thanks to its ability to capture share information among modalities.

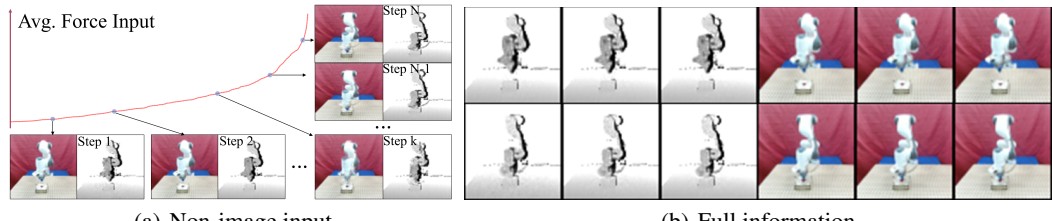

(a) Non-image input                                         (b) Full information

Figure 7: Learning to reconstruct visual event in robotics scenario. (a) image sampled from SMVAE conditioned on force and action inputs. SMVAE is able to reconstruct different visual modalities that depicts the relative position of the robotic arms to the target box properly. (b) From top to bottom, sampled reconstructed images from a action sequence

### 4.4 CASE STUDY: ROBOTICS CONTROL APPLICATION

The second case study shows that our method is readily applicable in robotics control scenarios using Vision&Touch datasetLiang et al. (2021). We use the SMVAE to learn cross-modality generation from continuous sensory input to images. Emerging human-in-the-loop shared autonomy systems are often equipped with multiple sensors, which pose a high requirement to the model's ability to learn co-representationLee et al. (2020); Luo et al. (2021); Chen et al. (2021); Selvaggio et al. (2021); Newman et al. (2022); Li et al. (2021). The Vision&Touch dataset is a real-world robot manipulation dataset that contains visual, tactile, control action, and robot proprioception data which pocess more diverse modalities. The robotic arm attempts to insert the peg located on its tip into the target object. We use a total of 4 modalities including the depth images, RGB images, the 6-axis force sensor feedbacks, and the control action given to the robotics arm in each time step. Fig. 7(a) illustrates that as the robotic arm is not receiving force signals in early steps, reconstruction results of the RGB image show clearly that the arm has no contact with the taget box below. Only when the robotic arm is receiving high force readings, the generated image depicts the contact between the robotics arm and the target box. The quality of the reconstructed rgb and depth images is also differ between partial observation and full observation. While only limited information is observed (i.e., force and action inputs), our method is only able to reconstruct rgb and depth images that can properly reflex the relative posistion between the robotic arm and the target object (Fig. 7(a)). But when more information is presented, the latent variables can have more comprehensive information about the event and better reconstruction result as we removed the conditional independence assumption (Fig. 7(b)).

## 5 CONCLUSION

This paper proposes a multimodal generative model by incorporating the set representation learning in the VAE framework. Unlike the previous multimodal VAE methods, the proposed SMVAE method provides a scalable solution for multimodal data of variable size and permutations. Critically, our model learns the joint posterior distribution directly without additional assumptions for factorization, yielding a more informative objective and the ability to achieve co-representation between modalities. Statistical and visualization results demonstrate that our method excels with other state-of-the-art multimodal VAE methods. Which has high potential in emerging multimodal tasks that need to learn co-representation of diverse data sources while taking missing modality problems or set-input processing problems into consideration. Application on cross-modality reconstruction in robotic dataset further indicates the proposed SMVAE has high potential in emerging multimodal tasks. In the future, we will explore methods that extend the current SMVAE framework to more diverse modalities as well as dynamic multimodal sequences to provide solutions for real-world multimodal applications.

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
