# OpenReview forum: "Generaling Multimodal Variational Methods to Sets"
_ICLR.cc/2023/Conference — Submitted to ICLR 2023_

### Official Review · Reviewer_LHYz · 2022-10-19

**Confidence:** 5
**Correctness:** 1
**Technical Novelty And Significance:** 2
**Empirical Novelty And Significance:** 1
**Recommendation:** 1

**Clarity, Quality, Novelty And Reproducibility:**

This paper requires additional work to improve clarity, mathematical rigor, and reproducibility (the key novelty of the paper is the use of self-attention, which is not clearly explained).

There is some merit in terms of novelty, as the key idea of this work is to generalize even further than what has been done in the literature, the function to obtain the joint approximate posterior using self-attention. The problem is that it is not clear how this can work.


**Strength And Weaknesses:**

The strength are:
* This work addresses an important topic in representation learning

The weaknesses are:
* The editorial quality must be drastically improved, both on clarity of exposition, and on mathematical rigor.

* Several aspects of the mathematical formulation of the proposed method are elusive.
The parametrization of the prior with parameters $\theta$ is not what you actually do. The prior has no parameters in the remaining of the paper.
Eq.1 has a typo.
The distinction between embedding and encoder networks is not clear.
Eq.3 has a typo.
Training process is not clear: do you assume to have all input modalities perfectly aligned, but omit some of them at training time?
The introduction of function $g(\cdot)$ in eq.7 is confusing. Are we talking about the encoder network?
The “noise outsourced functional representation” is not clear: I assume this is the latent variable.
The way the multi-head attention mechanism is used is obscure: keys, values and queries are not clearly specified.
The deterministic set representation function is indicated again by the function $g(\cdot)$, which is confusing.

* The section on total correlation is (to the best of my understanding) not correct. A recent article explicitly tackling multimodal representation learning with a total correlation objective is, for example, [1]


[1] @inproceedings{hwang2021-neurips,
    title={Multi-View Representation Learning via Total Correlation Objective},
    author={HyeongJoo Hwang and Geon-Hyeong Kim and Seunghoon Hong and Kee-Eung Kim},
    booktitle={Advances in Neural Information Processing Systems},
    editor={A. Beygelzimer and Y. Dauphin and P. Liang and J. Wortman Vaughan},
    year={2021},
    url={https://openreview.net/forum?id=SV4NhqUoO8}
}

* The experimental section is very weak, especially compared to the literature on the topic, including some of the references cited in the paper, such as Shi et al. ‘19. The datasets are simplistic, compared to the current state of the art, and the metrics used to compare methods are insufficient, see for example Shi et al. ‘19 and subsequent work from the authors, as well as Sutter et al. 20, and subsequent work from the same group.


**Summary Of The Paper:**

The work presented in this paper targets multimodal representation learning, following a line of work that uses a variational autoencoder (VAE) as a basis that is extended to learn a latent representation from multiple modalities, that is observations of the same phenomenon through the lenses of a variety of input sources. The objective is to build a model that scales to a potentially large number of input modalities, and that is capable of reconstructing the inputs, even in the case of missing modalities.

The training objective is an extension of the traditional VAE ELBO, that considers subsets of input modalities at training time. Instead of using either product or mixture of experts (or combinations thereof) to compute the joint posterior distribution, the authors propose to use a multi-head self attention layer. The claim is that the proposed approach does not require the assumption of conditional independence of a modalities given the latent variable, thus allowing to learn correlations between input modalities, which are encoded in the latent representation.

A series of experiments on synthetic multimodal datasets, that extend classical datasets such as MNIST, FashionMNIST and CelebA to account for a one-hot-encoded of the available labels as additional modalities, the authors compare the proposed method to alternatives from the literature (MVAE, MMVAE and JMVAE), using likelihood and conditional variants.
Finally, the authors discuss two real-life use cases, providing visual examples of the benefit of the proposed method.


**Summary Of The Review:**

This is a preliminary work that requires additional work to be understood by a reader, additional mathematical rigor and clarity of exposition. The experimental section is not on par with current literature, and should be extended by considering more realistic and challenging multimodal datasets, as well as appropriate evaluation metrics.

---

### Official Review · Reviewer_PMPV · 2022-10-20

**Confidence:** 4
**Correctness:** 2
**Technical Novelty And Significance:** 3
**Empirical Novelty And Significance:** 2
**Recommendation:** 3

**Clarity, Quality, Novelty And Reproducibility:**

There is a serious lack of clarity and quality. Sections 3.2 and 3.3 are particularly hard to understand and are riddled with typos. The objective should be stated explicitly and the theoretical results would benefit from a clearer statement of the necessary assumptions and the relevance and novelty of the results. The empirical results should be more thorough: the evaluation lacks a comparison with previous methods on more realistic multimodal datasets, and there are no ablation studies for relevant components and hyperparameters.

The contributions are potentially significant and somewhat new. Intuitively, the benefit of a learned aggregation function is clear and somewhat novel (at least with respect to multimodal VAEs; see Strengths and Weaknesses). However, the claims about the drawbacks of existing methods need stronger theoretical and empirical support. Similarly, the claims regarding the outstanding performance compared to existing methods requires more thorough experiments.

The lack of clarity implies limited reproducibility as the implementation would not be straightforward given the information provided in the text. There are no standard deviations or uncertainty estimates for the experiments. However, it is positive that the authors provide code for their experiments.

**Strength And Weaknesses:**

**Strengths**

The authors propose a simple yet intuitive and potentially impactful idea: to use a learned aggregation function using an attention mechanism to model the joint posterior in multimodal VAEs.

The qualitative and quantitative results on bi-modal image/label datasets look promising.

The two case studies show promising results for real-world applications.


**Weaknesses**

The authors claim that posteriors with basic aggregation functions (e.g., PoE and MoE) lead to a "defective bound" for the optimization and a loss of semantic information. However, the paper provides insufficient theoretical or empirical evidence to back up these claims.

The term "co-representation" does not seem to be defined, neither in the manuscript nor in the cited works. The term is used throughout the manuscript, including in central places like the abstract, where it is stated that "learning the co-representation [...] is a long-standing endeavor".

Empirically, the SMVAE shows slight improvements in multimodal generative learning compared to some of the existing methods. However, the quantitative comparison only features very simple multimodal datasets with only two modalities, one of which represents one-hot encoded labels. It would be helpful, if the authors would compare the performance on slightly more realistic multimodal datasets, such as MNIST/SVHN, CUB, and PolyMNIST.

There are many components and hyperparameters whose effects on the performance are not sufficiently clear and should be studied with additional ablations. For example: L2-normalization, the aggregation vector in Equation (8), the use of multiple attention heads, no use of positional embeddings, etc.

There are several things unclear about the objective, which does not even seem to be defined explicitly. At training time, does the input to the multi-head attention function (Eq. 8) consists of two times the same subset of modalities? Do these two sets differ in their order of modalities? Is the objective a valid ELBO on the complete set of modalities?

Where is the proof of Theorem 1? Is it an existing result from previous work?

The related work section does not make it sufficiently clear how the proposed idea of using attention to model the correlations between modalities relates to existing multimodal models with attention mechanisms. E.g.:
- Arsha Nagrani, Shan Yang, Anurag Arnab, Aren Jansen, Cordelia Schmid, Chen Sun: Attention Bottlenecks for Multimodal Fusion. NeurIPS 2021
- Andrew Jaegle, Felix Gimeno, Andy Brock, Oriol Vinyals, Andrew Zisserman, João Carreira: Perceiver: General Perception with Iterative Attention. ICML 2021
- Xinyang Geng, Hao Liu, Lisa Lee, Dale Schuurams, Sergey Levine, Pieter Abbeel: Multimodal Masked Autoencoders Learn Transferable Representations. CoRR abs/2205.14204 (2022)

**Summary Of The Paper:**

The paper introduces the Set Multimodal VAE (SMVAE), a new type of multimodal VAE that uses self-attention to model the joint posterior as a learnable function of an arbitrary subset of input modalities. While previous approaches use basic aggregation functions (e.g., the product of experts) to aggregate the unimodal embeddings across modalities, the SMVAE employs a self-attention module, i.e., a learnable aggregation function. The paper claims that previous approaches are limited by the conditional independence assumption implied by the respective aggregation function that they use and that the self-attention aggregation is better for modeling correlations between modalities. Empirically, the SMVAE shows slight improvements over some of the existing methods as well as promising results in two case studies.

**Summary Of The Review:**

The paper proposes a simple yet intuitive and potentially impactful idea. However, there are strong claims that are not sufficiently well supported by the results provided. The paper has the potential to provide a solid contribution to the lineage of multimodal VAEs, but it requires a major revision. In its current form, I tend to reject the paper.

---

### Official Review · Reviewer_YcXY · 2022-10-24

**Confidence:** 4
**Correctness:** 2
**Technical Novelty And Significance:** 3
**Empirical Novelty And Significance:** 2
**Recommendation:** 3

**Clarity, Quality, Novelty And Reproducibility:**

## Clarity
The proposed work lacks clarity in some parts. It is not clear to me what makes the work perform well and not all terms are correctly defined (see Weaknesses and Questions).

## Quality
The paper lacks quality with respect to evaluation and a clear description of all building blocks. The idea of using the set function for multimodal data is interesting.

## Novelty
The paper presents a novel application of the set-based method to multimodal VAEs, which from a novelty point-of-view is enough.

## Reproducibility
The authors provide details for all architectures and networks used. I am missing details of the embedding sizes and other hyperparameters and their sensitivity.

**Strength And Weaknesses:**

## Strengths
- **Idea**: I find the idea of a set-based approach to multimodal learning interesting
- **Motivation**:  the idea is well-motivated including related work.

## Weaknesses
- **Evaluation**: Previous Work (MMVAE, MoPoE) has used datasets with more than 2 modalities, which is an additional level of difficulty, but highlights the sensitivity of different methods to the number of modalities. MMVAE and MoPoE (and others) have highlighted the trade-off between the quality of generated samples (can be measured using test set log-likelihoods), coherence of samples (could be measured in conditional coherence), and a meaningful latent representation (accuracy of latent representation classification). The evaluation performed in this paper seems limited with only reporting quantitative results for log-likelihoods, but not the other metrics.
- **Clarity**: Not all terms are well-defined. For instance, I could not find a definition of co-representation. Co-representation seems to be an important term because it is repeatedly used. But without giving a clear definition of it. It is unclear to me where the performance gains come from (Table 1). Is it the set-based formulation? Or the additional conditional total correlation term? In my opinion, it would strengthen the paper if more insights are given into what parts of the proposed method are responsible for the performance improvements, and how sensitive the method to hyperparameters is (see Questions)

## Questions
- How is $ p(\mathbb{X}|z)$ calculated? What is the cost of that?
- Does the proposed framework also work with simpler set functions? The objective in eq 10 does not define the set function. Hence, it should work with any function that fulfills theorem 1. Are there any insights on that?
- Are there additional hyperparameters needed to tune the optimization of eq. 10? If yes, are there any sensitivity analyses with respect to hyperparameter selection?
- At the end of section 3.2., the authors mention the use of additional L2-regularization. What is the effect of this regularization? To which term is it exactly applied? What is the regularization weight? It would make the final loss function of the method clear if more details would be provided.
- What is a co-representation? To me, the term is not clear, and I could not find a definition in this submission.
- In Section 3.3. the authors say that former multimodal VAEs are not able to learn co-representation. Is there any proof or reference for that? Or at least empirical evidence?

**Summary Of The Paper:**

The authors propose a multimodal VAE, which includes an intermediate set representation.
This fixed-size representation of every modality is then used for the mapping to the variational approximation.
The authors show the performance of their proposed method on a computer vision study, a label-image dataset, and a robotics application.

**Summary Of The Review:**

To me, this paper is not ready to be published at ICLR. I like the idea of using a set-based approach to multimodal VAEs, but the weaknesses are dominating the good idea.

---

### Official Review · Reviewer_y89s · 2022-10-25

**Confidence:** 5
**Correctness:** 3
**Technical Novelty And Significance:** 1
**Empirical Novelty And Significance:** 2
**Recommendation:** 5

**Clarity, Quality, Novelty And Reproducibility:**

Minor questions:
- Evaluation metrics. In Table 1, the quantitative results are not readily interpretable. Instead, [1] proposes several metrics which are more friendly. The authors are suggested to use the new evaluation metrics for easy reading.
- Typos. Page 4 (3.2, last paragraph): “... regulate the parameter 0\mu … ” →  “... regulate the parameter \mu …”

**Strength And Weaknesses:**

Strength:

+ This paper aims to address the scalability issues, which are well-motivated and critical in multimodal representation learning (MRL).

+ The proposed method is technically feasible with intensive theoretical justification.

+ Extensive experiments on multiple datasets empirically validate the performance of proposed techniques, especially by comparing them with existing related works.

Weaknesses:
- Incremental technical contribution. Leveraging self-attention to model cross-modal interaction is a well-studied idea in MRL empirically and literally.

- Computation overhead. In high-modality cases (e.g., >10 modalities), calculating self-attention is computationally prohibitive. It seems that the proposed method does not take into account this computational issue, which significantly compromises the claimed addressing of the scalability issues.

- Missing details. What is the function of I (the query vector) in Eqn.8? Is it to obtain the mean and variance of the joint distribution?
It is a common practice to add modality-type embedding to the input data. Does your method follow the same practice?
What is the purpose of adding noise to variance (Eqn.6)?
What is the function of I (the query vector) in Eqn.8? Is it to obtain the mean and variance of the joint distribution?
How many self-attention layers are used in your method?

**Summary Of The Paper:**

This paper studied multimodal VAE. Compared to existing methods that rely on the conditional independence assumption, this paper relaxes such assumption by introducing Set Multimodal VAE (SMVAE). The author conducted experiments on three multimodal datasets to verify the effectiveness of SMVAE.

**Summary Of The Review:**

My major concern is incremental technical contributions and missing computational cost analysis.

---

> ### Author Response · Authors · 2022-11-06
> **Could you provide the paper containing more interpretable metrics?**
>
> I cannot see the paper referenced as '[1]' in your review, could you provide the name of the paper in the comment?

---

### Decision · Program_Chairs · 2023-01-20

**Decision:**

Reject

**Justification For Why Not Higher Score:**

Reviewers share similar concerns.

**Justification For Why Not Lower Score:**

N/A

**Metareview: Summary, Strengths And Weaknesses:**

The paper introduces a new type of multimodal VAE that uses self-attention to model the joint posterior as a learnable function of an arbitrary subset of input modalities based on self-attention.

Strengths

The idea makes intuitive sense, as the modalities may not be equally similar to each other and the self-attention aggregation shall be more powerful than simpler aggregation methods.

Weaknesses

Reviewers have serious concerns on the presentation being not clear, and that some claims lack theoretical or empirical evidence to back up.